# Former Foodstuff Products (FFPs) as Circular Feed: Types of Packaging Remnants and Methods for Their Detection

**Alice Luciano [1,*], Sharon Mazzoleni [1], Matteo Ottoboni [1], Marco Tretola [1,2], Rosalba Calvini [3], Alessandro Ulrici [3], Michele Manoni [1], Cristian E. M. Bernardi [1] and Luciano Pinotti [1,4]**

[1] Department of Veterinary Medicine and Animal Sciences (DIVAS), University of Milan, 26900 Lodi, Italy; sharon.mazzoleni@unimi.it (S.M.); matteo.ottoboni@unimi.it (M.O.); marco.tretola@unimi.it (M.T.); michele.manoni@unimi.it (M.M.); cristian.bernardi@unimi.it (C.E.M.B.); luciano.pinotti@unimi.it (L.P.)

[2] Agroscope, 1725 Posieux, Switzerland

[3] Department of Life Sciences and Interdepartmental Centre BIOGEST-SITEIA, University of Modena and Reggio Emilia, 42122 Reggio Emilia, Italy; rosalba.calvini@unimore.it (R.C.); alessandro.ulrici@unimore.it (A.U.)

4   CRC I-WE (Coordinating Research Centre: Innovation for Well-Being and Environment), University of Milan; 20133 Milan, Italy
*   Correspondence: alice.luciano@unimi.it

**Abstract:** Alternative feed ingredients in farm animal diets are a sustainable option from several perspectives. Former food products (FFPs) provide an interesting case study, as they represent a way of converting food industry losses into ingredients for the feed industry. A key concern regarding FFPs is the possible packaging residues that can become part of the product, leading to potential contamination of the feed. Although the level of contamination has been reported as negligible, to ensure a good risk evaluation and assessment of the presence of packaging remnants in FFPs, several techniques have been proposed or are currently being studied, of which the main ones are summarized in this review. Accordingly visual inspections, computer vision (CV), multivariate image analysis (MIA), and electric nose (e-nose) are discussed. All the proposed methods work mainly by providing qualitative results, while further research is needed to quantify FFP-derived packaging remnants in feed and to evaluate feed safety as required by the food industries.

**Keywords:** former foodstuffs; circular economy; feed safety; microplastics; packaging remnants; stereomicroscope; computer vision; multivariate image analysis; electronic nose

## 1. Introduction

The growth in the global population underlies the great demand for food production. The large consumption of resources is associated with food production; in fact, approximately 30% of the earth's surface and 70% of water is used for growing crops, but the world's food waste has been estimated to be 44% of the dry mass of agricultural crops [1]. As reported by the FAO [2], the trend to consume more food creates a demand for producers to offer ready-made food or a prolonged shelf-life to induce volume purchases. This creates increasing challenges for sustainable agriculture and livestock production [3,4]. In the near future, the demand for animal protein will increase, and sustainable livestock farming will need to improve food security, nutrition and healthy diets, and animal health and welfare and address climate change issues [5]. In the past 60 years, animal diets have thus undergone substantial changes, especially regarding the use of alternative ingredients to limit the use of corn, wheat, and other standard cereals in favor of other biomasses such as former foodstuff products (FFPs).

According to the EU Regulation 2017/1017 [6], FFPs "are foodstuffs, other than catering reflux, which were manufactured for human consumption in full compliance with the EU food law but which are no longer intended for human consumption for practical or logistical reasons or due to problems of manufacturing or packaging defects or other defects and which do not present any health risks when used as feed." Sustainable feed and efficient nutrition strategies are thus required to reduce food losses, the production of $CO_2$, and the use of resources, such as land and water [1]. FFPs have high nutritional potential in terms of nutrients and energy content [7,8]. Their nutrient composition is comparable to cereals commonly used in animal nutrition, with the exception of fat content, which is usually higher in FFPs [9].

The main limiting factor of the use of FFPs in Europe is the lack of information on their nutritional properties and their safe use in animal diets [9]. From a circular economy perspective, by using FFPs, it is possible to reduce food losses since these ingredients are suitable for animal feed, especially for pigs, poultry, and young animals [4,9–11]. The transformation of FFPs into animal feed ingredients does not always guarantee the complete elimination of food packaging; thus, small packaging remnants can end up being ground together with the food [12].

The two main roles of food packaging are to contain food and to protect food products from the environment and damage. Other functions include providing consumers with information on ingredients and nutritional data, traceability, convenience and making the products tamper-free (i.e., the product cannot be touched or modified without the packaging being broken). Generally, the packaging sector provides two main types of materials, i.e., flexible and rigid. Rigid packaging includes glass, rigid metal/aluminum, and wood [2]. Flexible packaging includes materials such as film, foil, or paper sheeting and now tends to be used more frequently than rigid packaging.

In food production, packaging materials are used to ensure the maintenance of food quality and safety during transport and storage before reaching consumers [13,14].

To meet all these requirements, the packaging material industry offers not only industrial one-component materials but also composites combining different materials. These composites bring together the properties of the individual components to create a synergetic effect, such as an increased barrier between food and the environment. In addition, by combining different layers of various materials, a notably reduced material input can be achieved for a given volume of foods to be packed. Therefore, composite packaging systems reduce the volume and weight of packaging waste [15].

However, it is not always possible to separate and remove small packaging remnants from food. These packaging remnants remain in the final products and thus in the animal feed. As reported elsewhere [9,16], the main concern regarding the safety of former food products is food packaging remnants. With different technological processes, the feed industry routinely removes the packaging from food material during processing in dedicated plants. These processes differ depending on the starting food product.

As reported in Figure 1, in the case of bakery products, the processes involved are: milling, the use of air, drying and the use of blown air to remove the remaining packaging materials (plastic and paper), magnets to remove ferrous metals, and eddy current separation (ECS) to remove nonferrous metals.

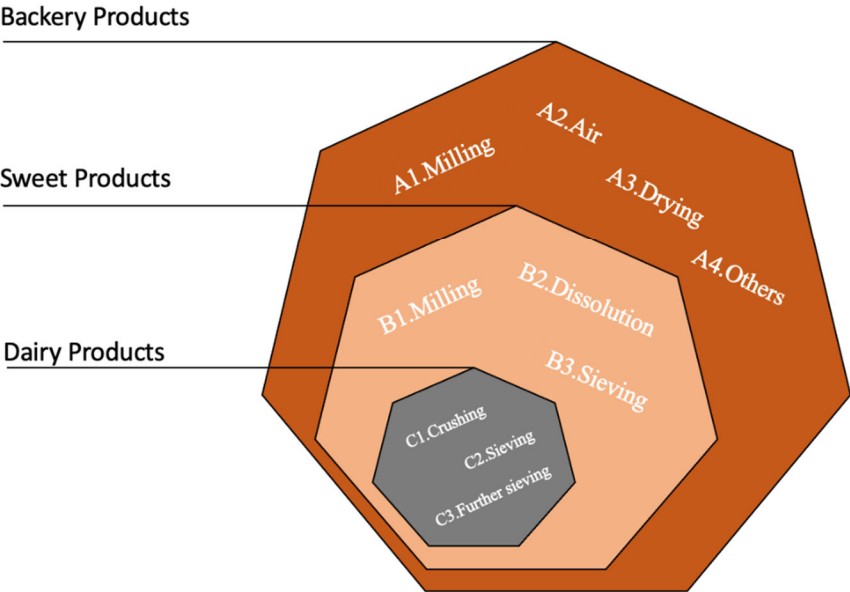

**Figure 1.** A1, coarse milling; A2, removal of most of the packing material with blown air or sieving; A3, drying (if necessary); A4, removal of the remaining packaging materials (plastic and paper) with blown air, ferrous metals with magnets, or nonferrous metals with an eddy current separator; B1.

coarse milling; B2, dissolution in water; B3, sieving; C1, crushing (i.e., crushing plastic yogurt cups/bottles); C2, sieving; C3, further sieving or centrifugation (if necessary).

ECS works by exposing conductive, nonferrous particles to a time-varying magnetic field, which in turn, gives rise to electrical currents throughout their volume [17]. The relative motion between the current and the magnetic field gives rise to a force called the Lorentz force, which subsequently deflects the nonferrous scrap remnants away from the nonmetallic fluff. The Lorentz force is the combination of an electric and magnetic force on a particle due to electromagnetic fields [17]. For sweet products, on the other hand, the technological processes consist of coarse milling, dissolution in water, and sieving. Finally, in the case of dairy products, the processes used are: crushing (e.g., squeezing plastic yogurt cups/bottles), sieving (packaging remnant removal), and centrifugation. The novelty of the present work is, thus, to provide an overview of the packaging remnants that are potentially present in former foods and to review the methods used for this purpose. There is limited information in the literature on the possible packaging residues that can become part of former food products, leading to the potential contamination of feed. There is even less information when the methods are considered. The present work, thus, aims to address the main types of packaging remnants that can be found in former foods and also the state-of-the-art methods available for their detection in feed. This review focuses on visual inspection [18], stereomicroscopy associated with computer vision (CV) [3], multivariate image analysis (MIA) [19], and electronic noses (e-nose) [4] and how they can be applied in feed and food quality and safety assessments.

## 2. Main Types of Packaging Remnants Found in Former Foods

Although exfood is nutritious and safe from a microbiological point of view [8,9], it can generate other safety issues, such as those related to packaging material. Packaging materials are not accepted as feed ingredients according to Regulation (EC) No. 767/2009 [20], which prohibits packaging materials from the agrifood industry for animal feeding to be placed on the market or used for animal nutritional purposes. The legal interpretation of this regulation is not clear, ranging from the prohibition of any remnant packaging material to the prohibition of intentional use. The packaging materials used with exfoods cover a large range of materials with often very complex compositions. The materials used for the packaging of human foods must comply with specific regulations. The European Regulation (EC) 1935/2004 [21] covers the general requirements for all types of packaging materials. It requires that packaging materials should not release their constituents at a level that could endanger human health. Specific EU directives have been published that regulate, in detail, the composition of plastics and regenerated cellulose. Other packaging materials (e.g., paper, coatings, or aluminum foil) are regulated in detail at the national level. However, as reported elsewhere [14], packaging mainly includes plastics, aluminum foil, cardboard and paper materials. These materials are the main types of packaging remnants found in former food products [4,18,22]. An appropriate approach is thus needed to define the right methods for detecting possible contaminants in exfoods used in the feed chain.

### 2.1. Plastic Materials

Today, plastics are very versatile materials with long-chain, low-cost, lightweight synthetic polymers that have numerous social benefits and have become a fundamental and apparently indispensable component of daily life [23]. Consequently, the global focus on managing plastic waste continues to intensify. In the literature, the presence of plastic remnants is reported mainly in seafood and water. In fact, high concentrations of plastic debris have been found in the oceans. This is caused by commercial shipping, fishing, and other activities, but also due to the increased release of micro- and nanoplastics through sewage or waste discharge caused by the increased use of plastic particles in cosmetics, textiles, fishing nets, packaging, and cleaning products. Many recent concerns, however,

have focused on microplastics. Since the first definition of microplastics by Thompson et al. [24], many studies have reported their presence in different marine environmental compartments, such as the Ross Sea (Antarctica), the Southern Portuguese Shelf, and the Atlantic Ocean [25–27]. In the case of marine environments, "microplastics" is a collective term that was first proposed by Thompson et al. [24], who defined the term as all plastic particles or debris smaller than 5 mm in diameter, including nanoplastics, according to the criteria of the US National Oceanic and Atmospheric Administration (NOAA) [28–30]. The classification of plastics is more complex. Specifically, plastic particles are known as mesoplastics (1–10 mm), microplastics (0.001–1 mm), and nanoplastics (<0.001 mm) in different aquatic environments [31–34]. In terrestrial environments, the recent literature has reported cases of the contamination of plastics in soil, soil earthworms, fruit and vegetables, and in the food chain generally [35–38]. The most common types of plastic in common use are polyethylene (PE; polythene) in a low-density form (LDPE; bin bags, film) and a high density construct (HDPE; shopping bags, bottle caps), terephthalate (PET; bottles, food trays), polypropylene (PPL; rigid tubs, straws), polyvinyl chloride (PVC; pipes, door and window frames), and polystyrene, both rigid (PS; food pots, toys) and expanded (EPS; packaging, insulation) [39]. Other plastic varieties exist, such as crystalline, or amorphous in a fluid matrix; however, these are usually for specialist usage. In terms of former foods, the main problem is the presence of packaging remnants. Of course, the marketing of feed ingredients containing packaging residues is prohibited; however, the bacterial load must also be contained below levels established by law to ensure animal well-being and health [9].

Figure 2 shows an example of the presumed plastic residuals found in a former food sample. They can be considered microplastics because their size is <5 mm.

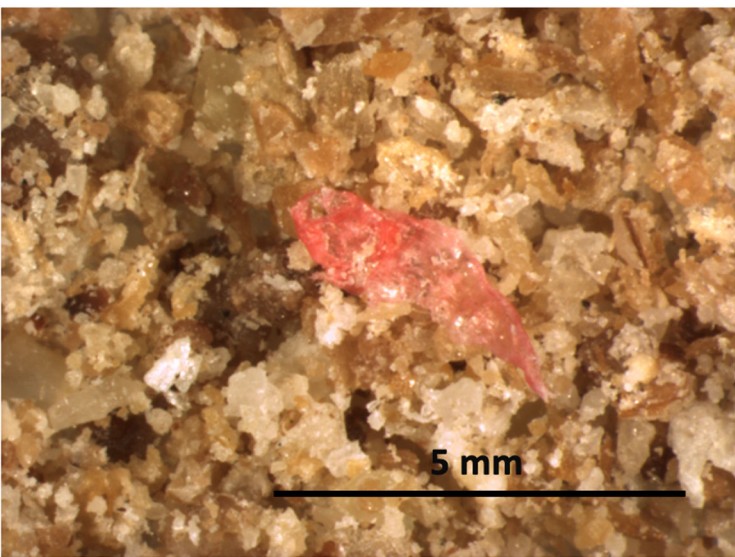

**Figure 2.** FPP sample and a piece of red/transparent packaging material (presumed to be plastic) obtained with a stereomicroscope and high-resolution CCD camera (CoolSNAP-Pro color camera).

These types of materials are not likely to be transported across cellular membranes, however, as they may be found in the gut. This explains why, in the case of specific feed of marine origin, such as fish hydrolysates, microplastics may be present. According to other authors [38], the major problem for food/feed safety is related to the presence of microplastics with a dimension of between 350 micrometers and 5 millimeters [40] that could originate from plastic fragmentation during processing [38]. As reported by the EFSA [41], although there is no literature available, microplastics may originate from other sources than the feed/food itself, e.g., processing aids, water, air, or released from machin-

ery, equipment and textiles. Moreover, it is possible that the number of microplastics increases during processing. The effects of other processes, e.g., pelleting, extrusion, cooking, and baking, on the content of plastics are still unknown.

### 2.2. Aluminum Materials

Aluminum (Al) is a natural element (the third most common element on earth) and, as a consequence, it is contained in different kinds of matrices, such as water, soil, and food [42,43]. Consumers increasingly want food that is ready to eat and storable for a few days or years; thus, foods need to be processed and packed in an optimal way, allowing an extended shelf-life, high security, and hygiene, and without changes in nutritive quality [15]. Aluminum is widely used in the food sector for packaging and containers and in direct contact with food. Aluminum foil plays an important role in modern food packaging [43,44]. It has different mechanical, physical, and chemical properties, such as a barrier effect and dead fold, and it can also legally come into contact with food. It thus has a wide range of applications in many different products and sectors [15,45,46]. Aluminum foil is light but strong and can be converted into complex shapes. It has excellent resistance to corrosion and high/low temperatures and shows high thermal and electrical conductivity [44,46]. Aluminum foil can be recycled without a decrease in quality [43]. Furthermore, aluminum foil packages are light and, thus, are energy efficient for transport. Aluminum foil meets consumer demand for packaging materials that combine functionality and environmental aspects. It is thus a durable packaging material for food, in particular for aseptic cartons, pouches (flat and self-standing), wrappings, bottle capsules, push-through blisters, laminated tubes, lids, trays, and containers [43,47]. The wide use of aluminum tools and foil, however, contributes to the increasing quantity of aluminum consumed through food [48]. The release of aluminum from packaging into foodstuff could represent a risk to human health [46]. Several studies have assessed aluminum release [43,49,50], as also addressed by the EFSA [51]. However, less is known about the methods of detection and characterization in food or feed of aluminum foil remnants. Below is an example of presumed aluminum residuals (Figure 3) in an FFP for animal feed.

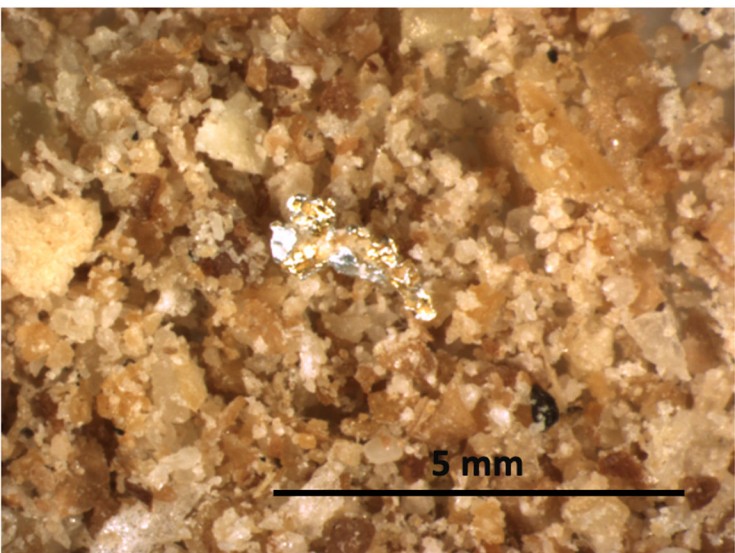

**Figure 3.** FPP sample and a piece of presumed aluminum packaging material obtained with a stereomicroscope and high-resolution CCD camera (CoolSNAP-Pro color camera).

### 2.3. Paper and Board Materials

Paper and board are also commonly used for food, particularly as they are very versatile. They are used in a wide range of containers, packaging materials, and food contact applications; for example, cups, dishes, paper towels, food boxes, tea bags, baking papers,

filters, beverage cartons, sacks, packaging for dry and frozen foods, including transport and distribution packaging, and tissue products [52]. Paper packaging can be made of parchment paper, or it can be used as bags to package loose foods. Carton board is commonly used for liquids and dry foods, frozen foods, and fast food. Corrugated board is widely used in direct contact with food (e.g., pizza boxes) and as secondary packaging [53]. Fibers originating from wood or from paper for recycling can be used in the manufacturing of pulp for food contact paper and board [54]. Paper and board are made of natural fibers of bleached or unbleached cellulose or are, alternatively, recycled from recovered materials. Chemical additives are needed in the manufacture of paper and board to achieve different technical functionalities. They are either added to the pulp during production or coated onto the surface afterward. Additives can be mainly categorized as functional additives or processing aids [55]. The first group of additives is used to modify the properties of the paper. They typically remain in the paper and include sizing agents, wet and dry strength resins, softeners, dyes, and pigments. Some toxicological studies on recycled paper have been published [54,56,57]. These include reports that recycled paper exhibits a higher in-vitro toxicity than virgin paper and that a toxicant or toxicants were identified. The paper and board industry has a longstanding commitment to the protection of human health and the interests of consumers through the provision of safe and functionally effective materials [52]. Again, less is known about the methods of detection and characterization in food or feed of paper and board remnants. Below is an example of a presumed paper remnant (Figure 4) in an FFP for animal feed.

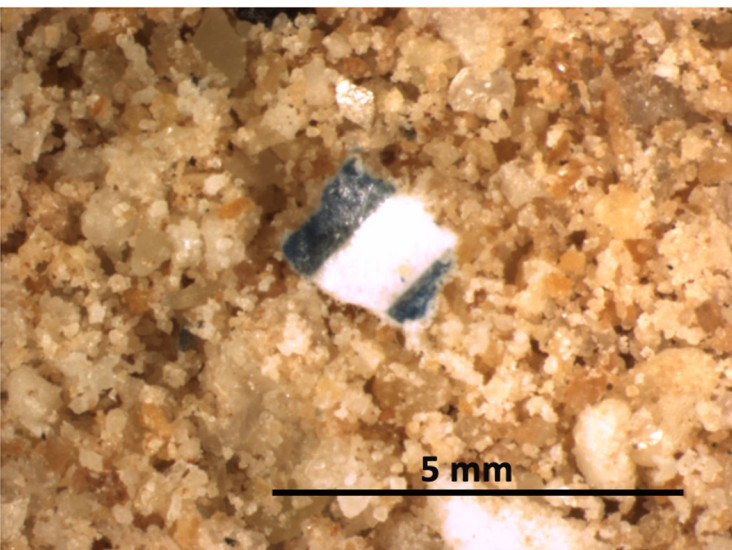

**Figure 4.** FPP sample and a piece of presumed white/green paper packaging material obtained with a stereomicroscope and high-resolution CCD camera (CoolSNAP-Pro color camera).

## 3. Methods for Detecting Packaging Remnants

### 3.1. Visual Inspection

Van Raamsdonk et al. [18,22] proposed a nonchemical and semidestructive method to detect and quantify packaging remnants in bakery products, including sweet bread and raisin bread. A visual inspection aims to detect and separate every particle that is considered by the operator as not native to the sample. This method is laborious and subjective because it depends on the ability of the operator to correctly recognize packaging remnants. The collected packaging remnants are then weighed, defatted, dehydrated, and finally weighed again [3,12,18]. In these preliminary studies, van Raamsdonk et al. [18,22] analyzed 243 samples, of which more than 90% showed a level of presumed contamination with remnants of packaging material under a level of 0.15% w/w. These particles were

defined as "presumed residuals" because it was not possible to identify the original packaging materials. The major problem involved in the proposed method was to confidently characterize these remnants in terms of their origin and nature. Some modifications were necessary, especially to the fraction of the matrix with particles smaller than 1 mm and to the cleaning of the particles of the packaging contaminants. The remnants detected by this visual method are usually bigger than 800 $\mu m$ [3,18]. Amato et al. [14] validated a fast and sensitive gravimetric method, based on the RIKILT method, for routine official controls to identify packaging residues in feed. A pelleted sample was sieved, each fraction was examined, and all the packaging materials were collected. Different parameters were used (specificity, limit of quantification, recovery, repeatability, reproducibility, and measurement uncertainty) [14]. In addition, to help the operator visually select the packaging remnants, stereomicroscopy was also used (Figure 2, Figure 3, Figure 4). Stereomicroscopy works on the low magnification observation of a sample, using light reflected from the surface of an object rather than transmitted through it. This instrument has been used in the food industry for inspection and quality control [58]. The results obtained by the visual inspection of FFPs, even with the use of a stereomicroscope, highly depend on the ability of the inspector to correctly recognize and quantify the different remnants [12]. Of these materials, paper, plastic, and microplastics are the most common. Microplastics are the most addressed contaminant in recent years [59], although toxicity assays that use concentrations over 100,000 times higher than those expected in the environment have limited practical relevance. Thus, adverse effects on animal and human health of current former food concentrations can be considered negligible [60]. Visual inspection can be unpredictable, time-consuming, and inconsistent [61]. Alternatives need to be found.

### 3.2. Computer Vision and Multivariate Image Analysis

Computer vision (CV) is performed with an instrument composed of a light chamber with a controlled white LED light equipped with a software-controlled CMOS camera able to obtain pictures in 16 million colors. The instrument is connected to software for system monitoring, data acquisition and multivariate statistics processing. Multivariate image analysis (MIA) is based on applying classical multivariate statistical methods in order to analyze images [62,63]. This methodology can work on images with more than one channel per pixel, for example, three red, green, and blue (RGB) channels in color images or spectral channels in multispectral and hyperspectral images. MIA can be used for classification, segmentation, defect detection, and even for predicting quantitative parameters [64].

Tretola et al., [3,12] demonstrated that CV could be a well-adaptable qualitative approach for detecting packaging remnants in FFPs. A key factor in this analysis was the white light condition [12]. In fact, the efficiency of CV strongly depends on optimal illumination conditions and intensity of light [62]. The standard protocol applied for the investigation of packaging remnants in FFPs using CV can be summarized as follows: (i) Pictures of FFP samples are taken using a high-resolution CDD (charge-coupled device) digital camera. (ii) The scanned image needs to be preprocessed before being analyzed in order to improve the image quality and details, after which (iii) the picture is divided into regions related to the areas of interest, and then (iv) the system uses statistical analysis and neural networks to obtain information on feed texture and grading [12]. Unfortunately, given the small size of packaging remnants in FFPs, the CV camera is not able to obtain pictures with a magnification that allows for good image analysis. Pictures of FFPs with higher magnification are thus needed using A stereomicroscope [3,12]. During image analysis, the CV captures the intensity of light in the red, green, and blue spectrum, obtaining information on the color of each pixel. For each picture, the color spectrum of the sample is represented by a histogram. Starting with the color spectrum derived from sample pictures, CV is able to formulate a statistical quality control chart, which includes conforming and nonconforming areas. RGB-based CV systems capture the intensity of the light in the red, green, and blue spectrum, obtaining information about the color of each

pixel. For each picture, the color features of the sample can be represented by a histogram. On the basis of the color features derived from sample pictures, a statistical quality control chart can be created, which includes conforming and nonconforming areas.

Since the analyzed sample has high variability, a training phase is required to relate the variability of the product to the sensor data recorded by the analysis system. At the end of the process, the software distinguishes between conforming training samples and the unknown sample [12]. Packaging remnants ground together with FFPs could have many different colors. For this reason, it is difficult to distinguish them from the background feed [3]. One strategy used to differentiate packaging remnants from the feed background is to evaluate the presence of a discriminant color, indicated by the software with a code, which is present only in the picture's pixels displaying packaging material but not feed. Comparing the color codes of each pixel with several pictures of standard FFP samples and FFP samples from which the packaging had been carefully removed using a stereomicroscope, the authors found a discriminant code that could be related to the specific presence of aluminum in the feed samples [12]. Therefore, based on the presence in the FFP color spectrum of a discriminant color code, CV is able to recognize the presence of packaging remnants (specifically aluminum) in pictures of contaminated FFP samples.

Calvini et al. [19] proposed an alternative strategy to detect packaging remnants based on MIA of RGB images of FFPs acquired by a stereomicroscope equipped with a digital camera. They used [19] six different commercial samples of FFPs from food companies in two different countries. MIA was applied following two different approaches, i.e., pixel-level analysis and image-level analysis. All samples included different food materials (broken biscuits, chocolates, croissants, bread, rice cakes, and breakfast cereals) and also contained particles of packaging remnants, consisting of paper, plastic, and aluminum. For image acquisition, aliquots of 5 g of FFP samples were placed in a petri dish, to form a single layer. A variable number of images (ranging from 5 to 13) was taken for each sample, considering sample aliquots both with and without packaging remnants. Firstly, every single image was analyzed at the pixel level using principal component analysis (PCA) in order to highlight similarities and differences among pixels related to the former food matrix and those related to the packaging remnants based on their color features. PCA was applied both to the RGB image "as is" as well as to the augmented RGB image obtained by considering additional color-related channels derived from the RGB values. In addition, the whole dataset of images was also analyzed at the image level, considering the colorgrams approach [65], which is a multivariate data dimensionality reduction method that allows the identification of outlier images of former food due to the presence of packaging particles. The results suggested that including additional color features derived from the RGB channels in the analysis allows to better highlight differences that are not clearly visible considering the RGB values alone, in particular when objects that need to be separated have similar colors. In practical scenarios, the development of specific models for different FFP types may lead to more accurate and reliable results. In light of the results obtained from the above-mentioned studies, CV, even when coupled with MIA, could be considered a faster qualitative screening approach, which simplifies the human effort in visual involvement. However, this approach is possible only when pictures of FFP samples are obtained by using a stereomicroscope and cannot be used to evaluate the presence of all kinds of packaging remnants [12].

### 3.3. Electronic Nose (e-nose)

The electronic nose (e-nose) is another recent, fast, and objective method to detect extraneous materials in both food and feed [4]. Through nonspecific chemical detectors, the e-nose simulates the olfactory system of humans and is used to identify and quantify simple and complex odors and aromas and also to discriminate between a wide range of odors [66]. These detectors interact with volatile organic compounds (VOCs) in the ana-

lyzed sample, and the output is an electronic signal. This signal originates from the interaction between semiselective sensors with VOCs and can be considered a fingerprint of the volatile molecules associated with the sample itself [62]. The system uses glass vials containing air with accumulated VOCs derived from each analyte, and through the use of a needle stacked in the cap of the vial, a gas sample is pumped to the e-nose sensors. These sensors correctly identify the presence of packaging materials in samples characterized by the same matrix, and, consequently, by the same volatile organic compound profile [4]. Basically, plastics, paper, and aluminum foil release their own volatile compounds, and the instrument uses VOC profiles as markers for detecting different concentrations of packaging remnants in the analyzed samples [4]. These results have shown that the e-nose is able to detect the presence/absence of packaging materials in FFP samples with the same matrix and the same VOC profile. The e-nose, therefore, distinguishes between clean and contaminated samples when they have the same odor background [4].

In fact, the presence of presumed packaging remnants is more reliable when the feed matrix has low variability (e.g., same batch, composed of the same ingredients, same odor prints etc.). It follows that packaging remnants can vary a great deal because they are treated with different inks for printing and solvents that can influence the sample odor profile. In these cases, the screening ability of the e-nose could be lower. At the same time, the results can also be explained by the limited quantity of presumed packaging remnants whose odor is covered by volatile compounds originating from the feed matrix [4]. The e-nose could thus be used to facilitate the activity of the stereomicroscope, thereby reducing working time and increasing the objectivity of the analysis [4]. The e-nose can thus be proposed as a modern analytical approach with large potential in addressing the authenticity, quality, and safety of food/feed and beverages [62].

## 4. Pros and Cons of the Methods Presented

The increasing focus on the safety of FFPs has led to the use of different methods aimed at the precise and effective detection of packaging materials. As reported in Table 1, using a stereomicroscope with a digital camera may not be totally effective and exhaustive. This method alone led to an underestimation of remnants that correlated with the laborious visual analysis by the operator.

**Table 1.** Main advantages and drawbacks of the various methods.

| METHODS | ADVANTAGES | DRAWBACKS |
|---|---|---|
| Visual inspection (Stereomicroscopy) | • Quantification<br>• Evaluation of heterogeneous distribution<br>• Partial identification of presumed packaging remnants origin<br>• Large sample size | • Underestimation<br>• Laborious/time consuming<br>• Operator dependent |
| Computer vision and multivariate image analysis | • Low-cost, fast, and objective analysis<br>• Easy differentiation between packaging residues color from FFPs matrix color<br>• Potentially feasible real-time image analysis<br>• Can be used in a wide range of new applications | • Optimal lighting conditions are necessary<br>• Needs to be used with stereomicroscope for a proper image resolution<br>• No determination of packaging remnants' nature considering only RGB color sensors |
| Electronic nose | • Great potential to discriminate experimentally cleaned samples from the standard and spiked samples | • Necessary to clarify the nature of the VOCs released by the packaging remnants<br>• Results affected by the feed matrix |

- No determination of packaging remnants' nature

To gain more precise and reliable results, it is possible to develop computer vision (CV) systems based on MIA of RGB images acquired with a stereomicroscope and a digital camera. Several strategies have been adopted to retrieve useful information from such images, thereby estimating the contamination of FFP samples more comprehensively in a short time with high accuracy.

A completely different method of analysis is the e-nose, which is an array of electronic chemical sensors with different selectivity patterns.

All these technologies are useful for a qualitative estimation of packaging remnants in FFPs.

However, the food and feed safety industries require methods of analysis that can quantify and characterize these remnants in feed, and further research needs to be focused on this objective. In light of this, future direction in studying potential methods for the detection of packaging materials in feeding stuffs should consider a combination of methods for detection, categorization, and quantification.

## 5. Conclusions

Former food products are an example of the circular economy. Reusing or recycling former food products as feed ingredients matches many national and international sustainability objectives in terms of reducing the use of limited natural resources and boosting resource efficiency. After FFPs have been collected, a new life cycle is initiated under the responsibility of the food chain operator, who has the obligation to ensure all uses of such material are in an adequately controlled way. Although former food is both nutritious and sustainable, it may be slightly contaminated with abiotic waste, such as paper, plastics, and aluminum foil, mainly coming from food packaging. This review has shown how these packaging remnants are present in FFPs as paper and board materials, plastic (micro and mesoplastic), and traces of aluminum foil. These materials are authorized as food packaging and, thus, should ensure a high level of safety. However, the safety levels must be constantly monitored through robust, accurate and—when possible—quantitative methods. None of the methods described in this review seem to meet all qualitative and, above all, quantitative requirements, and further investigation is needed to address this topic. Accordingly, a combination of methods is now needed to facilitate the detection, categorization, and quantification of packaging remnants in FFPs. However, the level of contamination of these former foods always seems to be low and in compliance with current European food legislation, in which all the member states agreed that "a zero tolerance for these traces is neither practical nor proportionate to the risk." For example, authorities in the Netherlands and Germany have undertaken their own risk assessments, and both countries now tolerate the presence of packaging up to a level of 0.15%. The UK Food Standard Agency also recommends a tolerance of 0.15% for such residues, while no specific limits have been set in EU legislation. Finally, even though these residues are present, the materials used as packaging for food intended for humans are, in any case, "generally recognized as safe" (GRAS) substances [13].

**Author Contributions:** Conceptualization, L.P., C.E.M.B., S.M. and A.L.; methodology, A.L., S.M., M.M., R.C., A.U. and M.O.; writing—original draft preparation, A.L., R.C., A.U., L.P. and S.M.; writing—review and editing, M.T.; supervision, L.P. and C.E.M.B.; funding acquisition, L.P. All authors have read and agreed to the published version of the manuscript.

**Funding:** The present work was conducted in the frame of the following projects: sustainable feed design applying circular economy principles: the case of former food in pig nutrition (SusFEED), funded by Fondazione Cariplo (Italy, Ref: 2018-0887); SAN-Progetto Grande Rilevanza Italia Serbia, funded by the Ministero degli Affari Esteri e della Cooperazione Internazionale; and the "One

Health Action Hub: University Task Force for the resilience of territorial ecosystems," funded by the Università degli Studi di Milano (PSR 2021-GSA-Linea 6).

**Data Availability Statement:** Not applicable.

**Acknowledgments:** The authors are grateful to the company that have provided the samples for the present work.

**Conflicts of Interest:** The authors declare no conflict of interest.

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
