# Peer review of "Former Foodstuff Products (FFPs) as Circular Feed: Types of Packaging Remnants and Methods for Their Detection"

_sustainability, doi:10.3390/su142113911_

Round 1

Reviewer 1 Report

The work is OK. However, some description is not scientific, for example, what is the scientific means for Figure 6?

Author Response

We are grateful to the reviewers for their comments.

REVIWER 1

The work is OK. However, some description is not scientific, for example, what is the scientific means for Figure 6?

The figure has been removed

Reviewer 2 Report

The present review article entitled “Former foodstuff products (FFPs) safety: methods for packaging remnants detection. An update” briefly discussed the various techniques such as machine vision and e-nose techniques to detect the contaminants in leftover foods. This is a unique paper which covers techniques for the detection of contaminants in FFPs and it will be helpful to both researchers and academicians. The paper contains several grammatical errors therefore; authors should seek help from native English speakers. However, before accepting the paper the authors should address the following issues

 1.      Authors should justify the novelty of their work in introduction section before the objectives.

2.      Fig 1: Need to cross-check it and what is meant by scrushing ? Dose dairy products require sieving?

3.      The authors clearly fail to differentiate between Machine vision techniques and multivariate image analysis sections. Even in the case of machine vision techniques RGB cameras can be used. However, in the case of machine vision multivariate and hyperspectral spectral cameras can be used.

4.      382-Thanks to this procedure ?-Check it

5.      In Table 1, some of the statements in the drawback section were wrong in the case of computer vision and multivariate image analysis-Verify it.

6.      A separate section should be included for future prospects

7.      The conclusion part is very poor it should be strengthened and avoid citing references in the conclusion section. The authors should draw conclusions from their study, not from other studies.

Author Response

We are grateful to the reviewers for their comments.

REVIWER 2

The present review article entitled “Former foodstuff products (FFPs) safety: methods for packaging remnants detection. An update” briefly discussed the various techniques such as machine vision and e-nose techniques to detect the contaminants in leftover foods. This is a unique paper which covers techniques for the detection of contaminants in FFPs and it will be helpful to both researchers and academicians. The paper contains several grammatical errors therefore; authors should seek help from native English speakers.

The paper has been checked by a mother tongue editor of academic English, and the certificate is attached.

However, before accepting the paper the authors should address the following issues

  1. Authors should justify the novelty of their work in introduction section before the objectives.

Done in lines 98-108.

  1. Fig 1: Need to cross-check it and what is meant by scrushing ? Dose dairy products require sieving?

This has been clarified in lines 88-89 and 97-98. The reason for sieving has also been added.

  1. The authors clearly fail to differentiate between Machine vision techniques and multivariate image analysis sections. Even in the case of machine vision techniques RGB cameras can be used. However, in the case of machine vision multivariate and hyperspectral spectral cameras can be used.

The sections Computer Vision and Multivariate Image Analysis (3.2) were merged as correctly suggested by the reviewer. In addition, also Table 1 has been modified accordingly.

  1. 382-Thanks to this procedure ?-Check it

Done

  1. In Table 1, some of the statements in the drawback section were wrong in the case of computer vision and multivariate image analysis-Verify it. 

Done

  1. A separate section should be included for future prospects

This has been added in the revised version line 415-418

  1. The conclusion part is very poor it should be strengthened and avoid citing references in the conclusion section. The authors should draw conclusions from their study, not from other studies. 

 The Conclusions have been revised.

Reviewer 3 Report

This review paper summarized the use of former foodstuff products as packaging materials, and their safety was also discussed. This writing is very good and easy to follow. I have only a few minor comments.

1) The title should be modified to make it clearer.

2) Fig.5 is not clear, can you provide a version of this figure with much higher resolution?

3) In some places, there are some language errors, please check and revise.

Author Response

We are grateful to the reviewers for their comments.

REVIWER 3

This review paper summarized the use of former foodstuff products as packaging materials, and their safety was also discussed. This writing is very good and easy to follow. I have only a few minor comments.

  • The title should be modified to make it clearer.

Done

  • 5 is not clear, can you provide a version of this figure with much higher resolution?

It is an output of the software and so is the best available.

  • In some places, there are some language errors, please check and revise.

The paper has now been checked by a mother tongue editor in academic English and the certificate is attached.
